# Equivalence between the Existence of Best Proximity Points and Fixed Points for Some Classes of Proximal Contractions

**Sumit Som** [1], **Moosa Gabeleh** [2,*] **and Manuel De la Sen** [3]

[1]   Department of Mathematics, School of Basic and Applied Sciences, Adamas University, Barasat 700126, India
[2]   Department of Mathematics, Ayatollah Boroujerdi University, Boroujerd 68, Iran
[3]   Institute of Research and Development of Processes, University of the Basque Country, 48940 Leioa, Spain
**\***   Correspondence: gabeleh@abru.ac.ir or gab.moo@gmail.com

**Abstract:** In the year 2014, Almeida et al. introduced a new class of mappings, namely, contractions of Geraghty type. Additionally, in the year 2021, Beg et al. introduced the concept of generalized $F$-proximal contraction of the first kind and generalized $F$-proximal contraction of the second kind, respectively. After developing these concepts, authors mainly studied the best proximity points for these classes of mappings. In this short note, we prove that the problem of the existence of the best proximity points for the said classes of proximal contractions is equivalent to the corresponding fixed points problems.

**Keywords:** best proximity point; $F$-proximal contraction; $F$-contraction of Hardy-Rogers-type; generalized $F$-proximal contraction of the first kind; generalized $F$-proximal contraction of the second kind; Geraghty contraction

**MSC:** $47H10$; $54H25$

## 1. Introduction

In 2012, Wardowski [1] first introduced the notion of $F$-contraction mapping and proved the fixed point results of such mappings in the context of metric spaces. After that, Cosentino and Vetro [2] came up with the notion of $F$-contraction of Hardy–Rogers-type as a generalization of $F$-contraction mapping and showed the existence of fixed points for such a class of mappings. Cosentino and Vetro proved the following fixed point theorem in [2].

**Theorem 1.** *[2] Let $(X, d)$ be a complete metric space and $T : X \to X$ be a self mapping such that $T$ is an F-contraction of Hardy-Rogers-type with coefficients $a, b, c, e, \tau, L$. Then $T$ has a fixed point. Moreover, if $a + e + L \leq 1$ then $T$ has a unique fixed point.*

If the mapping is non-self, then it may not have fixed point(s). Best proximity point theory discusses the theory of best proximity point(s), when the mapping is non-self. In case of self mapping, best proximity point(s) are nothing but the fixed points. In the year 2021, Beg et al. [3] introduced new classes of non-self mappings and developed the theory of best proximity points for these new classes of mappings. For the definition of generalized $F$-proximal contraction of the first kind and second kind, approximatively compactness, $p$-property, readers can see [3] for details. In this paper, we mainly deal with the best proximity point results ([3], Theorems 3.1 and 3.2) and ([4], Theorem 3) . After proving ([4], Theorem 3), Almeida et al. stated the following fixed point theorem as a corollary.

**Corollary 1.** *([4], Corollary 3) Let $(X, d)$ be a complete metric space and $T : X \to X$ be a continuous contraction of Geraghty type. Then $T$ has a unique fixed point.*

In this paper, $\Xi$ denotes the class of functions $H : (0, \infty) \to [0, 1)$ such that $H(s_n) \to 1 \implies s_n \to 0$ as $n \to \infty$. This class of functions is used to define contraction of the Geraghty type in [4]. For details, one can see [4]. In this current paper, we show that the existence of best proximity point for these new classes of mappings can be achieved from the corresponding fixed point results (we refer to [5,6] for different approaches to the same problem). Throughout this article $\mathbb{R}^+$ denotes the set of all positive real numbers and $\mathbb{R}$ denotes the set of all real numbers. Let $f : \mathbb{R}^+ \to \mathbb{R}$ be a mapping satisfying the following conditions:

(i) $f$ is strictly increasing;
(ii) for every sequence $\{\alpha_n\}_{n \geq 1}$ in $\mathbb{R}^+$ we have, $\lim_{n \to \infty} \alpha_n = 0 \iff \lim_{n \to \infty} f(\alpha_n) = -\infty$;
(iii) there exists $s \in (0, 1)$ such that $\lim_{\alpha \to 0+} \alpha^s f(\alpha) = 0$.

Some examples for this class of functions can be found in the work of Wardowski and Beg et al. In this paper we denote this class by $\Omega$. For the definition of $F$-contraction and $F$-contraction of Hardy-Rogers-type, readers can see the paper of Wardowski, Cosentino and Vetro.

We need the following result from [7].

**Lemma 1.** *([7], proposition 3.3) Let $(A, B)$ be a non-empty and closed pair of subsets of a metric space $(X, d)$ such that $B$ is approximatively compact with respect to $A$. Then $A_0$ is closed.*

## 2. Main Results

**Theorem 2.** *([3], Theorem 3.1) is a straightforward consequence of Theorem 1.*

**Proof.** Let $x \in E_0$. As $P(E_0) \subseteq G_0$, we have $P(x) \in G_0$. So, there exists $y \in E_0$ such that $d(y, P(x)) = dist(E, G)$. We show that $y \in E_0$ is unique. Suppose there exists $y_1, y_2 \in E_0$ such that $d(y_1, P(x)) = dist(E, G)$ and $d(y_2, P(x)) = dist(E, G)$. So,

$$d(y_1, y_2) = d(Px, Px)$$
$$\implies d(y_1, y_2) = 0$$
$$\implies y_1 = y_2.$$

Define a mapping $S : E_0 \to E_0$ by $Sx = y$ having the property that $d(Sx, Px) = dist(E, G)$. Now, we show that $S$ is an $F$-contraction of Hardy–Rogers-type. Let $x_1, x_2 \in E_0$ with $S(x_1) \neq S(x_2)$. Now,

$$\tau + F(d(S(x_1), S(x_2))) \leq$$
$$F\Big(ad(x_1, x_2) + bd(S(x_1), x_1) + cd(S(x_2), x_2) + h(d(S(x_2), x_1) + d(S(x_1), x_2))\Big)$$
$$\implies \tau + F(d(S(x_1), S(x_2))) \leq$$
$$F\Big(ad(x_1, x_2) + bd(S(x_1), x_1) + cd(S(x_2), x_2) + hd(S(x_2), x_1) + hd(S(x_1), x_2)\Big).$$

This shows that the mapping $S : E_0 \to E_0$ is an $F$-contraction of Hardy–Rogers-type. Additionally, from Lemma 1, we can conclude that $E_0$ is closed. So, there exists $z \in E_0$ such that $S(z) = z$. Also, $d(z, Pz) = d(Sz, Pz) = dist(E, G)$. This shows that $z$ is a best proximity point for the mapping $P : E \to G$. The uniqueness of the best proximity point for the generalized $F$-proximal contraction of the first kind mapping $P$ has been shown by Beg et al. in [3], so it is omitted. □

**Theorem 3.** *([3], Theorem 3.2) is a straightforward consequence of Theorem 1.*

**Proof.** Let $x \in E_0$. Since $Px \in G_0$ so, there exists $u_x \in E_0$ such that $d(u_x, Px) = dist(E, G)$. Let

$$\mathbb{A}(Px) = \{h \in E_0 : d(h, Px) = dist(E, G)\}.$$

So, $\mathbb{A}(Px) \neq \phi$. We show that the set $\mathbb{A}(Px)$ is singleton for each $x \in E_0$. Let $w_x, z_x \in \mathbb{A}(Px)$. Then

$$d(w_x, Px) = dist(E, G) \text{ and } d(z_x, Px) = dist(E, G).$$

So,

$$d(w_x, z_x) = d(Px, Px) = 0$$

$$\implies w_x = z_x.$$

Let us define a function $\mathcal{F} : P(E_0) \to P(E_0)$ by $\mathcal{F}(Px) = Pu_x$ where $u_x \in \mathbb{A}(Px)$. Now, we show that the mapping $\mathcal{F} : P(E_0) \to P(E_0)$ is an $F$-contraction of Hardy–Rogers-type. Let $x_1, x_2 \in E_0$ with $\mathcal{F}(Px_1) \neq \mathcal{F}(Px_2)$. Since $Px_1, Px_2 \in P(E_0) \subseteq G_0$, so, there exists $u_1, u_2 \in E_0$ such that $d(u_1, Px_1) = dist(E, G)$ and $d(u_2, Px_2) = dist(E, G)$. So,

$$\tau + F(d(Pu_1), Pu_2))) \leq$$

$$F\Big(ad(Px_1, Px_2) + bd(Pu_1, Px_1) + cd(Pu_2, Px_2) + h(d(Pu_2, Px_1) + d(Pu_1, Px_2))\Big)$$

$$\implies \tau + F(d(\mathcal{F}Px_1, d(\mathcal{F}Px_2))) \leq$$

$$F\Big(ad(Px_1, Px_2) + bd(\mathcal{F}Px_1, Px_1) + cd(\mathcal{F}Px_2, Px_2) + h(d(\mathcal{F}Px_2, Px_1) + d(\mathcal{F}Px_1, Px_2))\Big).$$

Let $(x_n)_{n \geq 1}$ be a sequence in $E_0$ such that $Px_n \to u \in G$ as $n \to \infty$ for some $u \in G$. For each $n \geq 1$, there exists $v_n \in E_0$ such that $d(v_n, Px_n) = dist(E, G)$. It can be easily seen that $d(v_n, u) \to dist(E, G)$ as $n \to \infty$. Since $E$ is approximatively compact, there exists a subsequence $(v_{n_k})$ such that $v_{n_k} \to v_0$ as $k \to \infty$ for some $v_0 \in E$. Since $P : E \to G$ is continuous so, $Pv_{n_k} \to Pv_0$ as $k \to \infty$. Now, we show that this limit does not depend on the subsequence of the sequence $(v_n)$. Let $(v_{m_k})$ be another subsequence such that $Pv_{m_k} \to Pu_0$ as $k \to \infty$. Since $d(v_{n_k}, Px_{n_k}) = dist(E, G)$ and $d(v_{m_k}, Px_{m_k}) = dist(E, G)$, so we have

$$d(v_{n_k}, v_{m_k}) = d(Px_{n_k}, Px_{m_k}).$$

As $k \to \infty$ we have $d(v_0, u_0) = 0$ and $v_0 = u_0$. This shows that $\lim_{k \to \infty} Pv_{n_k} = \lim_{k \to \infty} Pv_{m_k}$. Similarly, it can be shown that if $(x_n)$ and $(y_n)$ be two sequences in $E_0$ such that $\lim_{n \to \infty} Px_n = \lim_{n \to \infty} Py_n$ then $\lim_{n \to \infty} Pv_n = \lim_{n \to \infty} Pw_n$ where $d(v_n, Px_n) = dist(E, G)$ and $d(w_n, Py_n) = dist(E, G)$. Let us define another function $\overline{\mathcal{F}} : \overline{P(E_0)} \to \overline{P(E_0)}$ by the following. Let $x \in \overline{P(E_0)}$. Then there exists a sequence $(x_n)$ in $E_0$ such that $Px_n \to x$ as $n \to \infty$. For each $n \geq 1$ there exists $v_n \in E_0$ such that $d(v_n, Px_n) = dist(E, G)$. We define,

$$\overline{\mathcal{F}}x = \lim_{k \to \infty} Pv_{n_k} = Pv_x$$

where $(v_{n_k})$ be a subsequence of $(v_n)$ with $v_{n_k} \to v_x$ and $Pv_{n_k} \to Pv_x$ as $k \to \infty$. Now, we show that the mapping $\overline{\mathcal{F}} : \overline{P(E_0)} \to \overline{P(E_0)}$ is an $F$-contraction of Hardy–Rogers-type. Let $q_1, q_2 \in \overline{P(E_0)}$ with $\overline{\mathcal{F}}q_1 \neq \overline{\mathcal{F}}q_2$. Then there exists two sequences $(x_n)$ and $(y_n)$ in $E_0$ such that $Px_n \to q_1$ and $Py_n \to q_2$ as $n \to \infty$. Let $(v_n)$ and $(w_n)$ be two sequences in $E_0$ such that

$$d(v_n, Px_n) = dist(E, G) \text{ and } d(w_n, Py_n) = dist(E, G)$$

for all $n \geq 1$. Then there exists two subsequences $(v_{n_k})$ and $(w_{m_k})$ such that $Pv_{n_k} \to \overline{\mathcal{F}}q_1$ and $Pw_{m_k} \to \overline{\mathcal{F}}q_2$ as $k \to \infty$. So,

$$\tau + F(d(Pv_{n_k}, Pw_{m_k})) \leq$$

$$F\Big(ad(Px_{n_k}, Py_{m_k}) + bd(Px_{n_k}, Pv_{n_k}) + cd(Py_{m_k}, Pw_{m_k}) + h(d(Px_{n_k}, Pw_{m_k}) + d(Py_{m_k}, Pv_{n_k}))\Big)$$

Since $F$ is increasing, so,

$$d(Pv_{n_k}, Pw_{m_k}) \leq$$

$$ad(Px_{n_k}, Py_{m_k}) + bd(Px_{n_k}, Pv_{n_k}) + cd(Py_{m_k}, Pw_{m_k}) + h(d(Px_{n_k}, Pw_{m_k}) + d(Py_{m_k}, Pv_{n_k}))$$

As $k \to \infty$ we have,

$$d(\overline{\mathcal{F}}q_1, \overline{\mathcal{F}}q_2) \leq$$
$$ad(q_1, q_2) + bd(q_1, \overline{\mathcal{F}}q_1) + cd(q_2, \overline{\mathcal{F}}q_2) + h(d(q_1, \overline{\mathcal{F}}q_2) + d(q_2, \overline{\mathcal{F}}q_1)).$$

Consequently, we have,

$$\tau + F(d(\overline{\mathcal{F}}q_1, \overline{\mathcal{F}}q_2)) \leq$$
$$F\Big(ad(q_1, q_2) + bd(q_1, \overline{\mathcal{F}}q_1) + cd(q_2, \overline{\mathcal{F}}q_2) + h(d(q_1, \overline{\mathcal{F}}q_2) + d(q_2, \overline{\mathcal{F}}q_1))\Big).$$

So, by Theorem 1, $\overline{\mathcal{F}}$ has a fixed point in $\overline{P(E_0)}$. First of all, let there exists $x^* \in P(E_0)$ be such that $\overline{\mathcal{F}}x^* = x^*$. Let $x^{**} \in E_0$ be such that $d(x^{**}, x^*) = dist(E, G)$. This implies $d(x^{**}, \overline{\mathcal{F}}x^*) = dist(E, G)$. So, $d(x^{**}, Px^{**}) = dist(E, G)$. In this case $x^{**} \in E_0$ is the best proximity point of $P : E \to G$. On the other case let, $q \in \overline{P(E_0)} - P(E_0)$ be such that $\overline{\mathcal{F}}q = q$. In this case, there exists $(x_n)$ in $E_0$ such that $Px_n \to q$ as $n \to \infty$. Let $(y_n)$ be a sequence in $E_0$ be such that $d(y_n, Px_n) = dist(E, G)$. Then by definition

$$\overline{\mathcal{F}}q = \lim_{k \to \infty} Py_{n_k} = Pv_0$$

where $(y_{n_k})$ be a subsequence of $(y_n)$ with $y_{n_k} \to v_0 \in E$ and $Py_{n_k} \to Pv_0$ as $k \to \infty$. Now, since $d(y_{n_k}, Px_{n_k}) = dist(E, G)$, as $k \to \infty$ we have

$$d(v_0, q) = dist(E, G)$$
$$\implies d(v_0, \overline{\mathcal{F}}q) = dist(E, G)$$
$$\implies d(v_0, Pv_0) = dist(E, G).$$

In this case $v_0 \in E$ is the best proximity point of the mapping $P : E \to G$. The uniqueness of best proximity point is shown by Beg et al. in [3]. $\square$

**Theorem 4.** *([3], Theorem 3.3) is a straightforward consequence of Theorem 1.*

**Proof.** In ([3], Theorem 3.3), since the pair $(E, G)$ satisfies the $p$-property and $P(E_0) \subseteq G_0$, so, it can be seen that $E_0$ is closed. The proof is similar to Theorem 2, so omitted. $\square$

**Example 1.** *We apply our result (Theorem 3) to ([3], Example 3.4) to validate our claim. We will construct our function $\overline{\mathcal{F}} : \overline{P(E_0)} \to \overline{P(E_0)}$. In this case $P(E_0) = [0, \frac{1}{2}) \times \{1\}$ and $\overline{P(E_0)} = [0, \frac{1}{2}] \times \{1\}$. The mapping $\overline{\mathcal{F}} : \overline{P(E_0)} \to \overline{P(E_0)}$ be defined by*

$$\overline{\mathcal{F}}(x, 1) = \begin{cases} (\frac{x}{2(x+2)}, 1) \text{ if } x \in [0, \frac{1}{2}); \\ (\frac{1}{10}, 1) \text{ if } x = \frac{1}{2} \end{cases} \tag{1}$$

*and, $(0, 1)$ is the fixed point of the mapping $\overline{\mathcal{F}} : \overline{P(E_0)} \to \overline{P(E_0)}$. Since $(0, 1) \in P(E_0)$ so, by our result $x^* = (0, 0)$ is the best proximity point.*

**Theorem 5.** *([4], Theorem 3) is a straightforward consequence of Corollary 1.*

**Proof.** Let $x \in A_0$. As $T(A_0) \subseteq B_0$, so, $T(x) \in B_0$. So, there exists $y \in A_0$ such that $d(y, T(x)) = dist(A, B)$. Now, we show that $y \in A_0$ is unique. Suppose there exists $y_1, y_2 \in A_0$ such that $d(y_1, T(x)) = dist(A, B)$ and $d(y_2, T(x)) = dist(A, B)$. So,

$$d(y_1, y_2) = d(Tx, Tx)$$
$$\implies d(y_1, y_2) = 0$$
$$\implies y_1 = y_2.$$

Define a mapping $S : A_0 \to A_0$ by $Sx = y$ having the property that $d(Sx, Tx) = dist(A, B)$. It can be seen that $A_0$ is a closed subset of $X$. We show that $S : A_0 \to A_0$ is a contraction of Geraghty type. Let $x_1, x_2 \in A_0$. Since $d(S(x_1), T(x_1)) = dist(A, B)$ and $d(S(x_2), T(x_2)) = dist(A, B)$, so, we have $d(S(x_1), S(x_2)) = d(T(x_1), T(x_2))$. Since $T : A \to B$ is a contraction of Geraghty type, so, there exists $\alpha \in \Xi$ such that

$$d(T(x_1), T(x_2)) \leq \alpha(d(x_1, x_2)) \max \{d(x_1, x_2), M(x_1, x_2) - dist(A, B)\}$$
$$\implies d(S(x_1), S(x_2)) \leq \alpha(d(x_1, x_2)) \max \{d(x_1, x_2), M(x_1, x_2) - dist(A, B)\}.$$

It can be easily seen that $S : A_0 \to A_0$ is continuous. So, from Corollary 1, there exists $z \in A_0$ such that $S(z) = z$. Moreover, $d(z, T(z)) = d(Sz, Tz) = dist(A, B)$. So, $z$ is a best proximity point for the mapping $T$. Uniqueness is shown by Almeida et al. in ([4], Theorem 3). $\square$

**Example 2.** *We apply Theorem 5 to ([4], Example 3) to validate our claim. In this example, $X = \mathbb{R}^2$ with usual metric, $A = \{0\} \times [0, \infty)$, $B = \{1\} \times [0, \infty)$. The mapping $T : A \to B$ be defined by*

$$T(0, x) = (1, \frac{x}{1 + x}), (0, x) \in A.$$

*In this case $A_0 = A$, and $B_0 = B$. We construct our function $S : A_0 \to A_0$ according to Theorem 5. Let $(0, x) \in A_0$. Let $(0, y) \in A_0$ be such that*

$$d\Big((0, y), T(0, x)\Big) = 1$$
$$\implies d\Big((0, y), (1, \tfrac{x}{1+x})\Big) = 1$$
$$\implies \sqrt{1 + (\tfrac{x}{1+x} - y)^2} = 1$$
$$\implies y = \tfrac{x}{1+x}.$$

*So, the mapping $S : A_0 \to A_0$ be defined by $S(0, x) = (0, \frac{x}{1+x})$. According to Theorem 5, $(0, 0)$ is the unique best proximity point of the mapping $T$.*

## 3. Conclusions

In [3], Beg et al. introduced the notion of generalized $F$-proximal contraction of the first kind of mapping, generalized $F$-proximal contraction of the second kind of mapping and presented a best proximity point result ([3], Theorems 3.1 and 3.2). In this paper, we show that the best proximity point results for generalized $F$-proximal contraction of the first kind as well as the second kind mappings can be achieved from ([2], Theorem 3.1). Additionally, the same kind of result is proved for a contraction of Geraghty type. Several examples are also discussed to validate our findings.

**Author Contributions:** Conceptualization, S.S., M.G. and M.D.L.S.; methodology, S.S., M.G. and M.D.L.S.; formal analysis, S.S., M.G. and M.D.L.S.; writing—original draft preparation, S.S., M.G. and M.D.L.S.; writing—review and editing, S.S., M.G. and M.D.L.S. All authors have read and agreed to the published version of the manuscript.

**Funding:** This research was funded by Basque Government (Grant No. 1207-19).

**Institutional Review Board Statement:** Not applicable.

**Informed Consent Statement:** Not applicable.

**Data Availability Statement:** Not applicable.

**Acknowledgments:** We like to thank the learned referees for giving several valuable suggestions which have improved the presentation of the paper. The third author is thankful for the support of Basque Government (Grant No. 1207-19).

**Conflicts of Interest:** The authors declare no conflict of interest.

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
