# Peer review of "Equivalence between the Existence of Best Proximity Points and Fixed Points for Some Classes of Proximal Contractions"

_axioms, doi:10.3390/axioms11090468_

Round 1

Reviewer 1 Report

Recently, Beg et al. introduced two new classes of non-self mappings namely, generalized F-proximal contraction of the first kind and generalized F-proximal contraction of the second kind respectively. This paper aims to research the problem of the existence of best proximity points for the said class of proximal contractions is equivalent of the corresponding fixed points problems. The authors have provided interesting and quite valuable results and I find this work is important to those areas with closely related research interests and the manuscript is quite well written. So, the reviewer recommends this paper for the acceptance of publication in Axioms.

Author Response

Dear reviewer, 

We greatly appreciate your comments. 

Authors 

Reviewer 2 Report

In this article, the authors discussed that the problem of the existence of best proximity points for the said class of proximal contractions is equivalent of the corresponding fixed points problems. The authors presented theorems and corresponding proofs. The results are correct and interesting.

1.     The author should put forward the research highlights of the paper.

2.     Some examples can be added to the paper.

Author Response

(The authors gave the same response as above.)

Reviewer 3 Report

The aim of this manuscript is the study of "Equivalence between the existence of best proximity points and fixed points for some classes of proximal contractions" and the goal of this manuscript is to resolve the problem of the existence of best proximity points for contraction of Geraghty type and to study the existence and uniqueness of best proximity points for such class of contractions.

The paper is organized as follows.

Section 1 is Introduction with a little description of the importance of the notions.

Section 2, Generalized F-proximal contraction of the first kind and second kind, is dedicated to  the definitions, there are needed to understand the new results. Hier there are given also the main results and an Example.

In Section 3 presented Contractions of the Geraghty type where are given another main results.

Section 4 contains the conclusions and Section 5 is dedicated to references.

I recommend to the authors to change the structure to the paper.  The manuscript should contain  Introduction,  Preliminaries, Main results, Conclusions,  References. I recommend also more examples.

The mathematical means are correct and sufficient detail is provided.

I recommend the publication of the manuscript after a short revision.

Author Response

(The authors gave the same response as above.)
